# A New Class of Synthetic Flavonolignan-Like Dimers: Still Few Molecules, but with Attractive Properties

**DOI:** 10.3390/molecules24010108

**Published:** 2018-12-29

**Authors:** Valeria Romanucci, Giovanni Di Fabio, Armando Zarrelli

**Affiliations:** Department of Chemical Sciences, University of Napoli ‘Federico II’, Via Cintia 4, I-80126 Napoli, Italy; valeria.romanucci@unina.it (V.R.); difabio@unina.it (G.D.F.)

**Keywords:** flavonolignans, milk thistle, *Silybum marianum*, silymarin, silibinin, silybin

## Abstract

In recent years, there has been increasing interest in dimeric molecules due to reports of their promising therapeutic value in the treatment of numerous diseases (such as cancer, HIV, Alzheimer’s and, malaria). Many reports in the literature have highlighted the ability of these molecules to interact not only with specific biologic receptors but also to induce a biological response that more than doubles the results of the corresponding monomeric counterpart. In this regard, flavonolignan dimers or simply *bi-*flavonolignans are an emerging class of dimeric compounds that unlike *bi*-flavonoids, which are very widespread in nature, consist of synthetic dimers of some flavonolignans isolated from the milk thistle *Silybum marianum* [L. Gaertn. (Asteraceae)]. This mini-review will discuss recent developments in the synthesis, characterization and antioxidant activity of new families of flavonolignan dimers, in light of emerging medicinal chemistry strategies.

## 1. Introduction

Flavonolignans are secondary metabolites present in a few plants (Asteraceae, Berberidaceae, Chenopodiaceae, Flacourtiaceae, Fabaceae, Poaceae, and the Scrophulariaceae species) commonly used in the human diet, of which consumption has increased considerably in recent years thanks to their beneficial biological activity [1]. Pelter and Haensel [2] first proposed the generic name "flavonolignan" for naturally occurring hybrid molecules that are biogenetically originated from ubiquitous flavonoids and monolignols (namely, *p*-coumaryl, coniferyl and, sinapyl alcohol) [3,4,5,6].

Flavonoid portions, the flavonolignans isolated from plants, may initially be divided into four types (Figure 1): flavanonol-, flavonol-, flavanone and flavone-type.

Flavonolignans have a broad structural diversity as a consequence of the C-C or C-O linkage of the C_6_C_3_ unit (monolignol unit) to the flavonoid nucleus in different positions, affording dioxane, furan, cyclohexane rings or simple ether side chains. In general, these compounds contain several chiral centers, hence they usually occur in nature into stereoisomer forms. By 2016, over forty flavonolignans, containing eight different flavonoid counterparts, had been isolated from over twenty species of plants [1].

The main source of flavonolignans is the milk thistle *Silybum marianum* (L. Gaertn. (Asteraceae)). A crude extract of its seeds, industrially manufactured and named silymarin [7,8], contains several flavonolignans such as silibinin [9] (as pair stereoisomers silybin A and silybin B), [10], isosilybin A and isosilybin B, silychristin A, 2,3-dehydrosylibin (A and B) [11] and silydianin (Figure 2). *S. marianum* enjoyed a period of high esteem in folk medicine as a liver tonic [12]. Its extracts were used to improve liver function, protect against liver damage and accelerate the regeneration of damaged liver cells. Clinical studies have confirmed the usefulness of standardized milk thistle extracts in cases of liver intoxication, cirrhosis and other chronic liver diseases related to alcohol abuse. Several decades ago, silibinin and other components of the silymarin complex were discovered to be beneficial in treating liver damage resulting from the consumption of toxic fungi, such as *Amanita phalloides* [13]. It has been recently demonstrated that silymarin inhibits the production of leukotriene which explains its anti-inflammatory effect and antibiotic action. Clinical tests confirm the positive effects found in the experimental studies [14].

The major component of silymarin is silibinin, which is an almost equimolar diasteromeric mixture of silybin A and silybin B, easily available due to its high proportion in silymarin (ca 30%). Isolation of both pure diastereomers in multigram quantities can be accomplished by a preparative HPLC method [10] or advanced chemo-enzymatic protocols [15,16]. In the past decade, the medicinal merits of silibinin have been extended to include the potential for preventing and treating various diseases, thanks to properties that include anti-cancer [17], neuroprotective [18,19], antiviral [20] and antifibrotic activities [21].

## 2. *Bi-*Flavonolignans: New Promising Synthetic Metabolite Dimers

*Bi*-flavonolignans are an emerging class of dimeric compounds that unlike *bi*-flavonoids, which are very widespread in nature, consists of synthetic dimers of some flavonolignans. Synthetic *bi-*flavonolignans have also been found to exhibit interesting antioxidant activities, at over twice the rate of their natural monomeric units. However, despite these attractive features, synthetic *bi-*flavonolignans represent a poorly explored compound family within drug discovery.

Even if the number of flavonolignan units available is naturally low, when coupled with the large number of possible permutations in the position and nature of inter-flavonolignan linkage, a considerable structural diversity within the *bi-*flavonolignan class could be provided. To date, there exist only a few different synthetic approaches and these typically generate compound collections that are based around a very limited range of structures.

Today the only *bi-*flavonolignans reported in the literature are characterized by two flavonolignan units conjoined through an alkyl, alkoxy or phosphodiester-based linker. The high structural diversity of the silymarin component (Figure 2) makes the design of chemical modifications a difficult but appealing challenge. Below, the synthesis of known *bi-*flavonolignans will be discussed, focusing on the molecular diversity and their resulting anti-oxidant properties.

### 2.1. Flavonolignan Dimers Linked by C-C and/or C-O Bridge

The first *bi-*flavonolignans were isolated only ten years ago from Gazek et al. [22,23]. In this paper, the authors reported that the oxidation of natural silibinin in the presence of a stable radical such as DPPH, prevents the evaluation of its radical scavenger mechanism of action due to the formation of complex mixtures of high molecular weight products that are very difficult to separate. In order to reduce the reactivity of natural silibinin, they studied methylated silibinin derivatives.

The methylation to hydroxyl at C-4′′ or C-7 (Figure 3) showed less reactivity than silibinin, about 30 and 80%, respectively. However, silibinin C-4′′ *O*-methylated only underwent a small slowdown of oxidative processes, giving a mixture of oligomers always too difficult to purify [23]. Instead, using silibinin *O*-methylated at C-7 and laccase-mediated oxidation with *Trametes pubescens*, *bi*-flavonolignans **1** and **2** (Figure 3) in a ratio of about 2:5 and with yields of 9% and 25%, respectively, were obtained [22]. The same products were obtained by an oxidation reaction based on K_3_Fe(CN)_6_ [23] with yields of 40 and 4%, respectively. It should be noted that the asymmetric products **1** and **2** were presumably a mixture of four diastereoisomeric products, while the symmetric product **3** was a mixture of three diasteroisomers. Interestingly, no significant results were obtained by applying the same catalyzed laccase reaction to the *O*-methylated at C-4′′ silibinin (overactive substrate). An extensive theoretical study [24] indicates that the site most susceptible to oxidative attack in the silibinin is the OH at C-4′′, which loses a hydrogen and give rise to a radical with high spin density on the oxygen O-4′′ and on the carbons C-1′′, C-3′′ and C-5′′. The dimerization process at ring E of the silibinin passes through the formation of a phenyl-phenyl link between the two radicals in the ketone-like form, which then undergoes tautomeric transformation to the enol form that is strongly favored by a thermodynamic point of view. The whole process is more facilitated in apolar solvents, since polar ones do not promote the formation of hydrogen bonds that stabilize the dimer form.

In the case of dehydrosilybin, both conformational and electronic point of views assumes that ring E is very similar to the corresponding ring of silibinin. The hydroxyl group at carbon C-3 is the site with the highest tendency to oxidation, with high spin density localized on O-3 and carbon C-3. If DHS is *O*-methylated at C-5, C-7, and C-4′′, the corresponding dimer **5** is obtained by DPPH treatment (Figure 4), the formation of which is thermodynamically favored in apolar solvents. It should be noted that the formation of the bond does not follow a tautomeric reorganization, but ring C of one of the two units, which loses flatness and has a simple bond between positions C-2 and C-3. Presumably, the reaction gives a mixture of diastereoisomers which differ in the configuration to the chiral centers C-2 and C-3 of the lower part, instead having configurations both *R* or both *S* at the C-7′′ and C-8′′ centers of the two flavonolignanic units (up to sixteen different compounds) [23].

The dehydrosilybins methylated to the hydroxyl groups at C-3 and C-7, in the presence of an oxidant such as DPPH, leads to the formation of **6** (a mixture of the two enantiomers and the *meso*-isomer) and **7** dimers (a mixture of the two pairs of enantiomers), with low yields and dependent on the solvent used. If the hydroxyl groups at C-3 and C-7 are free, a more abundant mixture is obtained but difficult to purify. The formation of such dimers had been correctly predicted by the theoretical study. It is important to emphasize that the dimers are not stabilized by conjugation but by the formation of intramolecular hydrogen bonds, partly counteracted by steric repulsion.

Gavezzotti et al. [25] obtained the dimers **2** starting from silybin A and silybin B, selectively benzylated to the OH group at C-7 (Figure 4), in a reaction catalyzed by a laccase from *Trametes versicolor*. After de-*O*-benzylation the dimers **4aa** and **4bb** were obtained in 11% and 7% yields, respectively. By the same reaction, the silydianin dimer **8** (Figure 5) starting from the unprotected corresponding flavonolignan, and the benzylated silychristine dimer **9** were also obtained [25].

Whereas for the silibinin a behavioural profile has been postulated, including the role of the various hydroxyl groups [26], it is not easy to explain the antioxidant behaviour of dimers and their relative differences. These certainly can’t only depend on a greater number of hydroxy/phenolic groups, but are probably to be found in the three-dimensional structures. In all cases the DPPH test shown a radical scavenger capacity better than the corresponding monomers. Moreover, DHS dimers are always better than silibinin dimers thanks to a double bound at C-2/C-3 which allows a broad conjugation between the rings A, B and C. It is worth noting the property of dimer **2**, which has an IC_50_ value almost five times higher than that of silibinin. Dimer **8** is active more than silydianin itself and almost as much as the Trolox^®^, a known reference antioxidant [25].

### 2.2. Flavonolignan Dimers Linked with Different Length Spacers

The first group of flavonolignan dimers, whose monomeric units are linked by a spacer between the monomeric units with different lengths and rigidity was proposed by Vavříková et al. in 2014 [27]. This paper reported the synthesis of *bi-*flavonolignans with dodecanediol acid esterified with hydroxyl groups at C-9′′ of the pure diastereoisomeric forms of silibinin, silybin A and silybin B (**10aa**, **10ab** and **10bb**, Figure 6) and dehydrosilybin **11**.

Products were obtained by a transesterification catalyzed by Novozym 435, a lipase B obtained from *Candida antarctica* immobilized on an acrylic resin. The length of the aliphatic linker between the two units of silybin was optimized: the short aliphatic chain did not allow the linkage of two monomeric units. On the other hand, by taking advantage of the significant higher reactivity of the OH group in the position C-7 through chemical synthesis, new dimers were prepared.

Classical reaction with *p-* or *m-*xylylene dibromide in the presence of K_2_CO_3_ was used to obtain dimers **12** and **13** with a more rigid xylenyl linker. Redox behavior and cytotoxicity on human umbilical vein endothelial cells, normal human adult keratinocytes, mouse fibroblasts (BALB/c 3T3) and human liver hepatocellular carcinoma cell lines (HepG2) were evaluated. In particular, the results obtained by DPPH and inhibition of microsomal lipoperoxidation assays reported that silybin dimers showed better activity than the monomers, on the contrary, the 2,3-dehydrosilybin dimer **11** presented weaker antioxidant/antilipoperoxidant activity than its monomer (DHS). Conversely, regarding the cytotoxicity aspect, the silybin dimers were more cytotoxic than the parent compound. Instead, the 2,3-dehydrosilybin dimer showed weaker cytotoxicity than the monomer [27].

Recently, a new family of flavonolignan dimers has been proposed by Romanucci et al. [28]: Phosphate-Linked Silibinin dimers (PLSd). Inspired by oligoflavonoid structures [29] and exploiting phosphoramidite chemistry [30,31,32,33], they reported an efficient synthetic strategy to obtain new silibinin dimers in which the monomeric units are linked through phosphodiester bonds (Figure 7).

Exploiting the selective protection of the hydroxyl groups of silibinin [34,35], the suitable building blocks (Figure 7) were synthesized in good yields. Silibinin 3-OH or 9′′-OH compounds were coupled with phosphoramidites building blocks in three different combinations, using 0.45 M DCI as the activator. After the oxidative treatment, deprotection from protecting groups and RP-18 HPLC analyses and purification, the products were converted into the corresponding sodium salts leading to the phosphodiester derivatives **14**–**16** (20%) in good yields (15–20%). PLSd have a good water solubility (more than 20 mgL^−1^) under circumneutral pH values, whereas the silibinin was found to be very poorly soluble (less than 0.4 mgL^−1^). Their antioxidant properties were evaluated by different approaches and preliminary antioxidant tests (DPPH test [36,37]) showed a pronounced increase in the activity of the dimers compared to to the silibinin. The anti-oxidant behavior of dimers **14**–**16** were also investigated towards two different ROS, ^1^O_2_ and OH radical. The ability to scavenge ^1^O_2_ in H_2_O was tested. Despite a similar reactivity for both dimers and silibinin, a higher reactivity towards the OH radical (about twice) was estimated for the dimers **14** and **16** with respect to the silibinin with a second order rate constant major than 1.5 × 10^10^ M^−1^·s^−1^. A value very similar to that reported for a known potent antioxidant as quercetin. The PLSd showed a very low toxicity towards HepG2 cells (even for long exposure times). The cytoprotective effect was evaluated on HepG2 cells and using the DCFH-DCF assay where the oxidative stress was carried out by Xanthine/Xanthine oxidase (X/XO) system. All the dimers showed a strong ability to scavenge most of the free radicals generated by X/XO on HepG2 cells and in all cases more efficiently than silibinin [29].

## 3. Conclusions

Dimer or oligomer molecules often exhibit enhanced biological activity relative to the simple monomer units and their pharmacokinetic parameters are improved [38,39]. Sometimes, an entirely new biological entity is created by dimerization or oligomerization of different building blocks [40,41,42,43,44]. Contrary to many flavonoids that exist naturally as dimer forms, the flavonolignan dimers are only synthetic. The source of flavonolignan monomeric units is the milk thistle *Silybum marianum* (L. Gaertn. (Asteraceae)), whose seed extract is named silymarin. In the last ten years, the synthetic approaches reported are different and range from chemical to enzymatic strategies. Initially, flavonolignan dimers have been observed in studies to understand the anti-oxidant (or pro-oxidant) activity of the different OH functions of flavonolignans as silibinin, silydianin or silychristin and theoretical studies have been conducted [22,23,24,25,26]. Starting from different *O*-alkylated (*O*-methyl or *O*-benzyl) flavonolignans, the reactivity profile of different OH groups was outlined. The focus of these studies was the reactivity of different OH groups in the presence of stable radical or enzymatic conditions and consequently, biological screening was not accomplished for dimers obtained.

In 2014, Vavříková et al. [27] reported the chemo-enzymatic synthesis of silibinin, silybin A and/or silybin B and 2,3-dehydro-silybin linked by different length spacers. This was one of the first syntheses starting from the two diastereoisomers, silybin A and silybin B [35]. Redox behavior and cytotoxicity on different line cells were evaluated and, in general, silibinin dimers showed better activity than the monomers, although the 2,3-dehydrosilybin dimer showed weaker antioxidant activity than its monomer. Conversely, the silibinin dimers were more cytotoxic than the parent compound. Instead, the 2,3-dehydrosilybin dimer showed weaker cytotoxicity than the monomer. It was recently reported that using inspiration from oligoflavonoid structures and the exploitation of phosphoramidite chemistry, resulted in an efficient synthetic strategy to obtain new silibinin dimers in which the monomeric units are linked through phosphodiester bonds: the Phosphate-Linked Silibinin dimers (PLSd) [28]. New dimers (PLSd) presented a good water solubility (more than 20 mg·L^−1^) under circumneutral pH values, whereas the silibinin was found to be poorly soluble (less than 0.4 mg·L^−1^). All dimers showed a strong ability to scavenge most of the radicals more efficiently than silibinin as well as a very low toxicity towards HepG2 cells (even for long exposure times). The combination of high antioxidant activity with low toxicity as well as their good water solubility, make the PLSd promising synthetic metabolites in the large class of polyphenols.

These last two strategies offer the possibility to design dimers with a high molecular diversity, both in the nature of the building blocks and the length of the spacer.

## Figures and Tables

**Figure 1 molecules-24-00108-f001:**
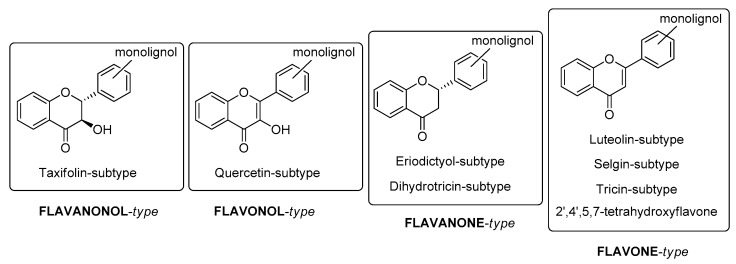
Classification of naturally occurring flavonolignans.

**Figure 2 molecules-24-00108-f002:**
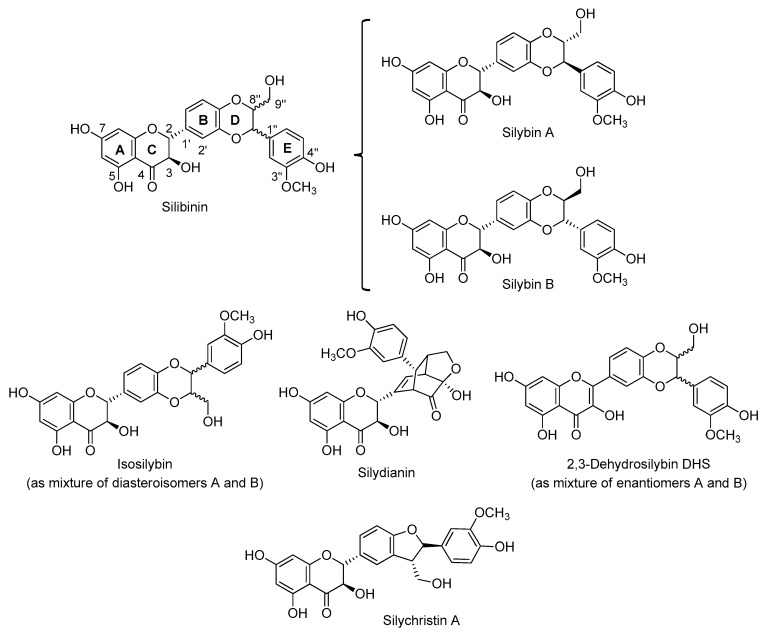
Main flavonolignans from *S. marianum* (L. Gaertn. (Asteraceae)).

**Figure 3 molecules-24-00108-f003:**
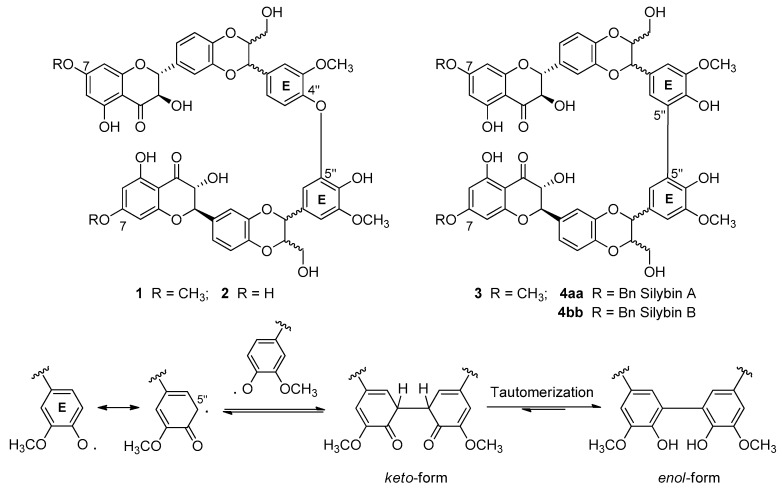
First *bi*-flavonolignans from silibinin oxidation and the mechanism of 5′′-5′′ bond formation.

**Figure 4 molecules-24-00108-f004:**
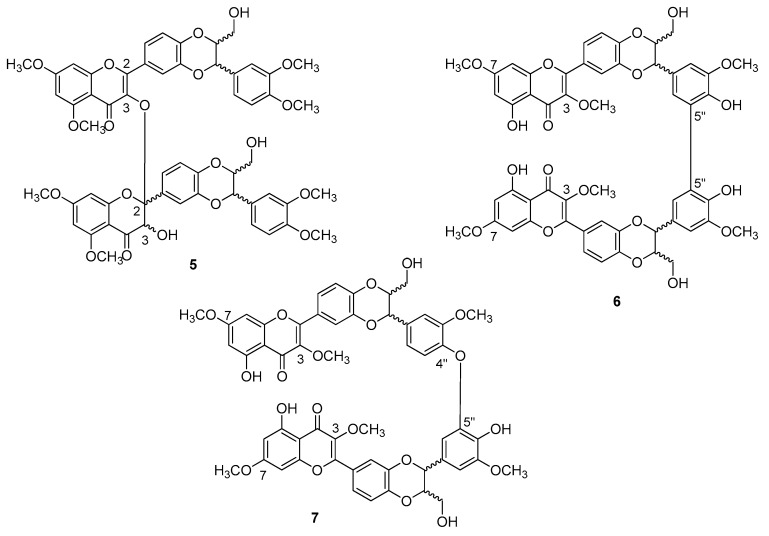
*Bi*-flavonolignans from dehydro-silybin oxidation.

**Figure 5 molecules-24-00108-f005:**
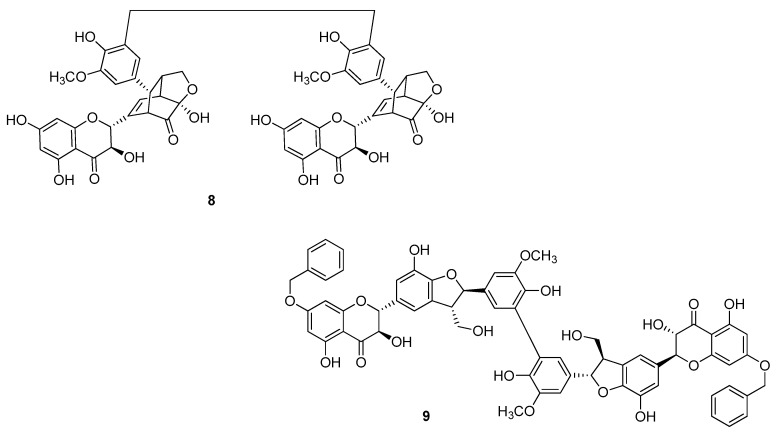
*Bi*-flavonolignans from silydianin (**8**) and silychristine (**9**) oxidation.

**Figure 6 molecules-24-00108-f006:**
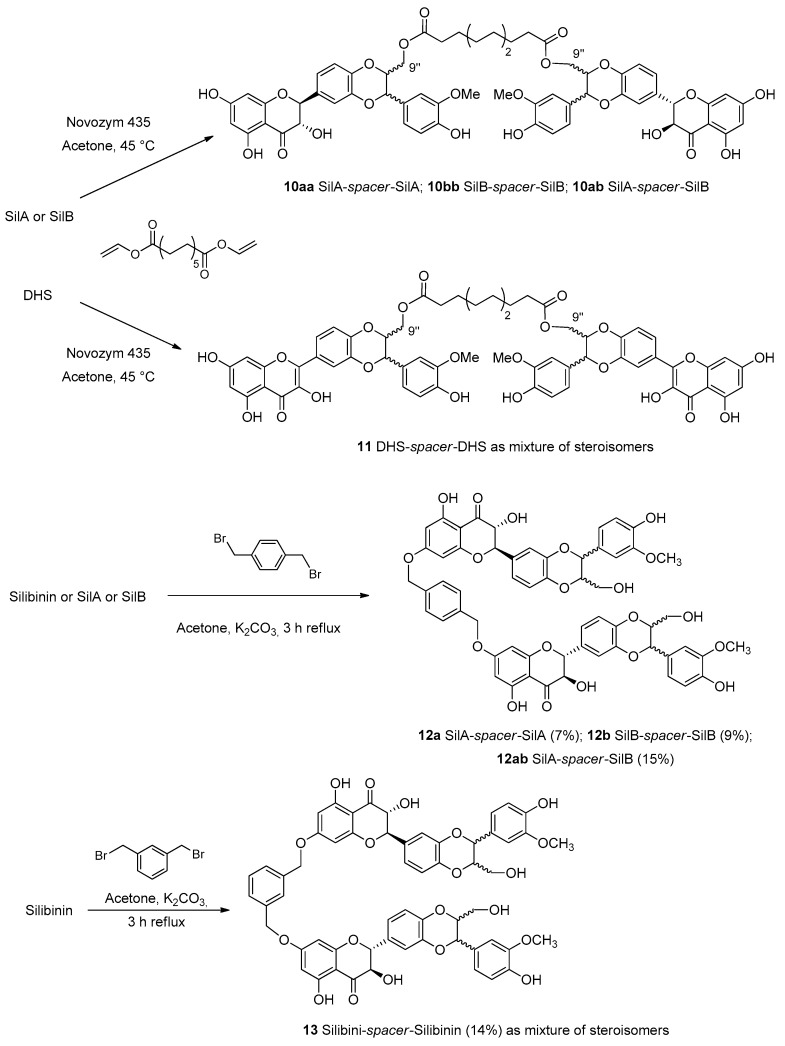
Lipase-catalyzed synthesis of silybin and dehydro-silybin (DHS) dimers (**10**–**11**) and synthesis of silibinin dimers with ether spacer (**12**–**13**).

**Figure 7 molecules-24-00108-f007:**
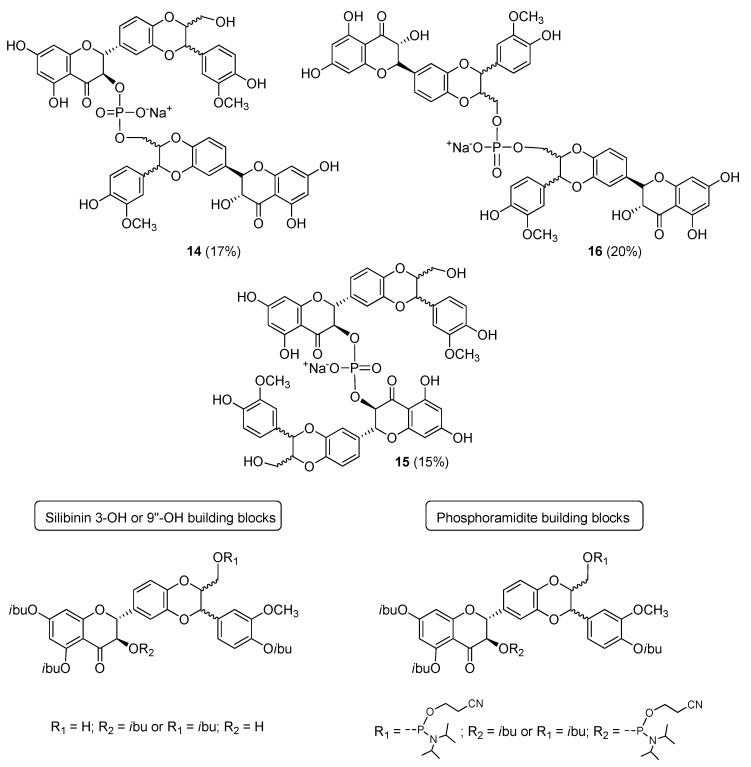
Phosphate-Linked Silibinin dimers (PLSd) and building blocks useful for their synthesis.

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
