# Peer review of "A New Class of Synthetic Flavonolignan-Like Dimers: Still Few Molecules, but with Attractive Properties"

_molecules, 2018, doi:10.3390/molecules24010108_

Round 1

Reviewer 1 Report

This mini-review entitled: A new class of synthetic flavonolignan-like dimers: still few but with attractive properties summarizes the most recent progress in the area of dimerized flavolignan. In the introduction part a brief description of flavonolignans is provided. Then the synthesis and bio-properties of C-C, C-O and other spacers is described. From technical point of view, the review is well organized possessing only several typo mistakes:

page 5, line 140: There is “Vavrikova” should be “Vavříková”

Scheme 13: There is “3h” should be “3 h”

Figure 7: Please explain what OCE means.

Since overall quality of the review is high and the topic covers attractive area from bio-medicinal chemistry I recommend to accept the review for publication in Molecules.

Author Response

Dear reviewer,

tanks for your suggestions.

Replay: All suggestions have been taken into due consideration and revisions have been made. Done (the correction are highlighted in yellow).

Best regards

Armando Zarrelli

Reviewer 2 Report

This review discusses the chemical properties of a new class of synthetic flavonoid compounds derived from the silymarin mixture extracted from Silybum marianum. The main weakness of this review is the lack of reference citations in the text, so a lack of references in the bibliography. The review is focused on the synthesis and the chemical properties of the new class of compounds. To improve the paper, the authors could add a paragraph on the possible use of this new class of synthetic flavonolignan dimers.

1.     Such a review should cite more references, in particular in the paragraphes ending in the following lines : 126, 137, 148, 160 and 185.

2.     Putative applications of these compounds could be discussed too.

3.     Please check spelling to improve English.

Author Response

Dear reviewer,

tanks for your suggestions.

Replay:  All suggestions have been taken into due consideration and revisions have been made. We have expanded the references by inserting some (highlighted in yellow). We have
expanded the discussion on the possible use of flavonolignans dimers and we have corrected
some words (highlighted in yellow).

Best regards

Armando Zarrelli

Reviewer 3 Report

The mini-review "A new class of synthetic flavonolignan-like dimers...." by A. Zarrelli et al. is an excellent review covering flavonolignan dimers as an merging class of compounds to exert a promising therapeutic values for the treament of numerous diseases, such as cancer, malaria, Alzheimer et.

The review is clearly written with many references including the most recent ones.

I recommend the publication of this review in Molecules.  

Author Response

Dear reviewer,

Tanks for your comments.

Best regards

Armando zarrelli